# Behavioral and Cognitive Comorbidities in Genetic Rat Models of Absence Epilepsy (Focusing on GAERS and WAG/Rij Rats)

**DOI:** 10.3390/biomedicines12010122

**Published:** 2024-01-07

**Authors:** Evgenia Sitnikova

**Affiliations:** Institute of Higher Nervous Activity and Neurophysiology, Russian Academy of Sciences, 5A Butlerova St., Moscow 117485, Russia; eu.sitnikova@ihna.ru

**Keywords:** genetic animal models, spontaneous absence epilepsy, drug-naive rats, fear-motivated learning, anxiety-like symptoms, depression-like symptoms, cognitive thalamus

## Abstract

Absence epilepsy is a non-convulsive type of epilepsy characterized by the sudden loss of awareness. It is associated with thalamo-cortical impairment, which may cause neuropsychiatric and neurocognitive problems. Rats with spontaneous absence-like seizures are widely used as in vivo genetic models for absence epilepsy; they display behavioral and cognitive problems similar to epilepsy in humans, such as genetic absence epilepsy rats from Strasbourg (GAERS) and Wistar Albino rats from Rijswijk (WAG/Rij). Both GAERS and WAG/Rij rats exhibited depression-like symptoms, but there is uncertainty regarding anxiety-related symptoms. Deficits in executive functions and memory impairment in WAG/Rij rats, i.e., cognitive comorbidities, are linked to the severity of epilepsy. Wistar rats can develop spontaneous seizures in adulthood, so caution is advised when using them as a control epileptic strain. This review discusses challenges in the field, such as putative high emotionality in genetically prone rats, sex differences in the expression of cognitive comorbidities, and predictors of cognitive problems or biomarkers of cognitive comorbidities in absence epilepsy, as well as the concept of “the cognitive thalamus”. The current knowledge of behavioral and cognitive comorbidities in drug-naive rats with spontaneous absence epilepsy is beneficial for understanding the pathophysiology of absence epilepsy, and for finding new treatment strategies.

## 1. Introduction

The International League Against Epilepsy and the International Bureau for Epilepsy define epilepsy as a neurological disorder characterized by an ongoing propensity to experience epileptic seizures and by the neurobiological, cognitive, psychological, and social implications of this condition [1,2,3]. Consequently, there is a considerable concern regarding the link between epilepsy and a range of neuropsychiatric comorbidities, such as anxiety, depression, and attention-deficit/hyperactivity disorder.

Absence epilepsy is a non-convulsive type of epilepsy characterized by the sudden temporary loss of awareness or consciousness [4,5]. Absence epilepsy is less common than the convulsive type of epilepsy, but it can be just as dangerous if not treated. In view of the fact that absence seizures can occur without warning, they can be extremely frightening for patients and their families [4,6,7,8].

Absence seizures are caused by excessive excitation and hyper-synchronization of the thalamocortical system [9,10,11,12,13,14,15]. This system is incredibly complex and huge, containing over a million neurons and with connections to over a hundred different areas of the brain [10,16,17,18]. Figure 1 shows the schema of the thalamo-cortical system. There are three major parts in the cortico-thalamo-cortical loop [17,19,20,21,22,23]: the pyramidal cells of the neocortex (Figure 1, green pyramids), which are excitatory neurons; the thalamocortical neurons of the relay and high-order thalamic nuclei, which are also excitatory neurons; the inhibitory neurons of the reticular thalamic nucleus. Cortical pyramidal neurons excite neurons in the relay thalamic nuclei (brown neurons in Figure 1) and inhibitory neurons of the reticular thalamic nucleus (blue neurons in Figure 1). The reticular thalamic nucleus intermittently inhibits thalamocortical cells. Thalamocortical cells provide excitatory feedback to the cortex and to the reticular thalamic nucleus [23,24]. Thus, the thalamocortical system generates sleep spindle oscillations and spike-wave discharges (SWDs) [10,12,21,22,25,26]. Dysfunction of the thalamocortical network could underlie cognitive comorbidities, as suggested by the concept of thalamocortical dysrhythmia [27,28] and the concept of the cognitive thalamus (the latter is introduced below, in Section 6).

SWDs are recognized as hallmarks of absence epilepsy in human and rat models [3,29,30,31,32]. The primary dysfunctions in information processing in the thalamocortical circuitry are associated with a variety of cognitive problems, such as executive function problems, attention deficit, thought problems, and many others [6,33,34,35,36]. Blumenfeld, in 2005, proposed that impaired consciousness in absence epilepsy may be caused by the focal disruption of information processing in specific corticothalamic networks, while other networks are spared [37].

A number of clinical studies have demonstrated that absence epilepsy is associated with some cognitive comorbidities, such as executive dysfunction, memory deficits, and impairment of visual-perceptual skills [6,34,35,36]. It is challenging to determine the effect of absence seizures on cognitive functioning, as the outcomes in various patient groups are inconsistent and may be influenced by medication. Therefore, genetic rat models with spontaneous absence seizures offer a valuable opportunity to better understand the neurocognitive comorbidities associated with absence seizures in humans.

Spontaneous absence seizures in genetic rat models could only be detected using electroencephalogram (EEG) through the presence of high-voltage spike-wave discharges (SWDs) [38,39,40]. The monitoring of EEG in freely moving rats requires chronically implanted electrodes, which is rather invasive. Earlier studies in genetically prone rats found almost no SWDs during cognitive tasks [41]. Only 4% of SWDs appear during active behavior [42]. Moreover, genetically prone rats were able to discriminate between meaningful and meaningless stimuli presented during SWDs [43]. Therefore, cognitive comorbidities could be considered consequences of a seizure disorder.

Some patients with absence epilepsy may have cognitive deficits even when they are seizure-free [29,44,45]. Attention improved during long-term follow-up in children with absence epilepsy, but other cognitive weaknesses persisted over time, regardless of seizure freedom [46]. Studies in drug-naive rats with a genetic predisposition to absence epilepsy could help to gain insight into the mechanisms of neurocognitive comorbidities associated with some anti-epileptic drugs. In the field of translational neuroscience, there is a serious problem with translating animal behavioral abnormalities into human neuropsychiatric comorbidities. Here, I am concentrated on cognitive and behavioral comorbidities in two widely accepted genetic rat models of absence epilepsy [31,32,39,47]: the inbred Wistar Albino Glaxo Rats from Rijswijk (WAG/Rij) and the Genetic Absence Epilepsy Rats from Strasbourg (GAERS). In respect to the terminology, the term “neurocognitive” seems to be too human-centric, but the term “neurobehavioral”—too animal-centric. Here I use the term “cognitive”, because it is more neutral and does not imply that only humans have brains and cognitive abilities.

## 2. Genetic Rat Models

In vivo animal models are essential for basic research on human diseases, preclinical studies, and the development of new medications [38,48,49,50,51,52]. There are two major types of animal models of neuropsychiatric disorders [48] and epilepsy: (1) the induced/experimental models, in which a pathological condition is induced through mechanical, chemical, or electrical means (e.g., kindling and pharmacological models of epilepsy); (2) the natural/genetic models, in which pathological processes are genetically predetermined, e.g., genetic rat models of absence epilepsy. Both types of animal models display behavioral and cognitive comorbidities. For the induced/experimental animal models of acquired epilepsy (i.e., the first type), neurobehavioral comorbidities have recently been systematically reviewed by W. Löscher and C. Stafstrom [53]. Here, I concentrate on the second type, genetic models of absence epilepsy and the accompanying behavioral and cognitive comorbidities.

Some rat strains are known to display inheritable disorders of the nervous system and are used as models for human diseases, like depression (Wistar–Kyoto rats [54,55,56,57] and Flinders Sensitive Line rats [58,59]), schizophrenia (Brattleboro rats, spontaneous hypertensive rats, apomorphine-susceptible rats [60]), and absence epilepsy (GAERS and WAG/Rij rats [32,39,47,61]). Belzung and Lemoine (2011) proposed a logical schema for validating animal models of psychiatric disorders and outlined five criteria for validity [62]: homological validity, pathogenic validity, mechanistic validity, face validity, and predictive validity. As early as 1984, Willner introduced three criteria for model validity that are well accepted in the area of translational neurology: face validity, predictive validity, and construct validity [63]. “*Face validity [is] the phenomenological similarities between the model and the condition being modeled… Predictive validity concerns the success of predictions made from the model, and construct validity concerns its theoretical rationale*” [63]. Willner introduced these criteria in order *“to assess animal models of depression*”, and these criteria have been widely used to validate various animal models of human diseases [62,64,65], including absence epilepsy [32,38]. Willner defined the face and predictive validities in relation to the antidepressant effects of drugs. The current review concerns the construct validity of animal models of absence epilepsy. The construct validity is “*a theoretical rationale of animal models both at the level of a similarity of the behavioral and/or cognitive dysfunctional processes and at the level of a similarity of the etiology*” [62]. In order to ensure construct validity, rat models should demonstrate behavior and cognitive processes that are comparable to those in human disease.

GAERS and WAG/Rij rats are widely used as reliable animal models of absence epilepsy with behavioral and cognitive comorbidities. In 1986, van Luijtelaar and Coenen introduced WAG/Rij rats as a genetic rat model of absence epilepsy [30]. They reported electrophysiological and behavioral signs of absence epilepsy in these rats [32,66]. WAG/Rij rats were proposed as a model for studying the epileptogenesis of absence epilepsy [67] in conjunction with neurological and psychiatric comorbidities [68]. Many studies use Wistar rats as an a priori control non-epileptic strain, ignoring the presence of absence seizures in many adults. For example, at the age of 9–12 months, approximately 18% of male Wistar rats obtained from Stolbovaya breeding stock in Russia showed epileptic SWDs [69]. Absence seizures have also been detected in all (100%) Wistar rats bred at the animal facility of Biological Psychology in Radboud University (Nijmegen, The Netherlands) [70]. Vergnes et al., in 1982, reported that 24 out of 63 (38%) adult Wistar rats in their laboratory in Strasbourg (Laboratoire de Neurophysioiogie, Centre de Neurochimie, CNRS, Strasbourg, France) expressed spontaneous absence-like seizures [71]. Wistar rats obtained from L’Arbresle (France) were screened for the presence of spontaneous SWDs, and Marques-Carneiro et al., in 2014, reported spontaneous absence-like seizures in 9 out of 11 rats (81%) at adulthood [72]. At the Faculty of Medicine in University of São Paulo (Brazil), Wistar rats have been selectively bred for the presence of spontaneous absence epilepsy up to F9 generation [73]. It appears that the prevalence of spontaneous absence-like epileptic activity in Wistar rats differs among colonies. Marques-Carneiro et al., in 2014, used Wistar rats as a second control group in addition to the non-epileptic control (NECs) for studying anxiety-related behavior in GAERS [72]. Taylor et al., in 2019, found spontaneous SWDs in wild *Rattus norvegicus* and examined their electrographic, pharmacological, circadian, and behavioral features. SWDs in wild rats showed a highly similar profile to SWDs found in outbred rats (such as Brown Norway and Long Evance) and in inbred WAG/Rij rats [74]. In summary, it is crucial to confirm the non-epileptic phenotype of Wistar rats in order to ensure their suitability as controls in studies of epilepsy and related comorbidities.

A well-recognized genetic rat model of absence epilepsy, GAERS, has been selected from the Wistar strain. As early as 1982, Vergnes et al. reported that 6–12-month-old Wistar rats in their 20-year-old breeding colony in Strasbourg (France) spontaneously exhibited absence seizures [71] and started selectively breeding subjects with spontaneous SWDs, GAERS and control NEC strain. GAERS were derived from an outbred Wistar colony and displayed a 100% incidence of SWDs. NEC rats were also derived by selective inbreeding for the lack of absence seizures. Recently, McCafferty et al. (2023) demonstrated that the GAERS model was able to reproduce key fMRI and behavioral characteristics of consciousness-impairing human absence seizures [75]. These authors found a decrease in neuronal firing in both the cortex and thalamus during SWDs and considered this to be a major pathological factor. The results of this study could lead to a better understanding of consciousness and seizure mechanisms, which could have important clinical applications in the future.

## 3. Investigation of Behavioral and Cognitive Functions in Rats

Current research on genetic rat models has found that absence epilepsy associates with *behavioral comorbidities*, which are expressed as depression- and anxiety-like symptoms [67,72,76,77,78,79,80], and with *cognitive comorbidities*—poor learning abilities [77,79,81,82,83]. A variety of behavioral tests can be conducted on rats to evaluate their behavioral functions. Listed below are some tests that are commonly used (Figure 2): the Forced Swimming test, the Tail Suspension test (developed for mice, and less commonly used for rats), the Sucrose Preference test, the Light–Dark choice test, the Elevated Plus maze, and the Open Field test [57,65,76,77,78,84,85]. The novel objects recognition test, social recognition and social interaction tests, and various tests for memory and learning skills are reviewed elsewhere [49,54,60].

The Forced Swimming test, also known as Porsolt’s test, is the gold standard for screening antidepressant drugs in rats [84,86,87]. The rat is placed into a tank of water from which it cannot escape. It shifts between actively trying to escape and staying passive. In this test, the rat’s immobility is used as an indicator of depressive-like behavior (or “behavioral despair” in terms of Porsolt [87]). Additionally, the duration of the first episode of active swimming (i.e., “climbing”) and the overall duration of swimming are measured.

The Tail Suspension test, which was developed for mice [56,80] and adapted for rats [64,88,89], is a behavioral test used to measure an animal’s reaction to stressful situations. The rat is suspended by the tail from a lever, the movements of the animal being recorded for 6 min during which the animal shows periods of agitation and immobility. In this test, the rat either attempts to escape through active movements or remains passive, similar to what is seen in Porsolt’s test. The efforts to escape can be classified into three types: (1) running movements; (2) body torsion with attempts to catch the suspending bond; (3) body jerks. After several failed attempts, the subject finally stops moving and hangs motionless. The Tail Suspension test is analogous to Porsolt’s test and produces “behavioral despair” [88,89]. The benefits of the Tail Suspension test can be summarized as follows: (1) rats do not experience hypothermia; (2) the recording of rat behavior is more accurate than with Porsolt’s Test; (3) it is more sensitive to low doses of drugs and provides a more distinct dose-effect relationship.

The Sucrose Preference test is the most commonly used method for measuring pleasurable experiences and hedonic drivers of consumption as well as anhedonia [90,91]. Chronic stress reduces rats’ physiological preference for sweet solutions, which is thought to be analogous to the anhedonic behavior observed in depressed patients and indicative of a decreased rewarding effect of sweet tastes. The test protocol varies slightly, but all versions involve presenting a rat with a free choice between two bottles, one containing a 1–2% sucrose solution and the other containing plain drinking water. This test can sometimes give inconsistent results. Rats could be deprived of food and water before the experiment. The concentration of sucrose usually ranges between 0.5% and 2%, with the duration of the test varying from 15 min to 8 days [92].

The Light–Dark choice test is a widely used test to measure anxiety-like behavior [78,93]. Rodents naturally avoid brightly illuminated areas. This test measures rats’ avoidance of bright areas, as well as their response to stressful situations, such as a new environment and light. The test is usually performed in a box with two chambers: a small, dark one and a large, illuminated one. A rat is placed in the brightly illuminated compartment and allowed to move between the illuminated and dark compartments. Brightspace anxiety is measured by the first latency to enter the dark compartment and the total time spent there. The number of transitions between compartments decreases over time due to habituation, and the dynamics of transitions between compartments is used to compute the index of activity-exploration.

The Open Field test is a quick and simple way to evaluate locomotion, anxiety, and stereotypical behaviors such as grooming and rearing [94]. There is a lot of debate about the interpretation of behavior in the Open Field test due to uncertainty about the motivation of rats’ behavior. Some believe that it is fear-induced escape behavior rather than anxiety. Anxiety can be measured in situations with both positive and negative outcomes, but the central area in the Open Field test is illuminated and considered to be aversive (i.e., a negative outcome). Rats become more averse to the central zone as their level of anxiety increases.

The Elevated Plus Maze is an established test for anxiety in rodents that has been used for more than 30 years [95]. This is a black plastic arena in the shape of a plus with two opposite arms enclosed by walls and two open arms. The rat is placed on the central platform and is allowed to explore three zones of interest: closed arms, open arms, and the central area. Rats with high levels of anxiety-like behavior tend to spend less time in the open arms of the plus maze.

## 4. Cognitive Comorbidities in WAG/Rij Rats

Over the past 10 years, the prevalence of cognitive comorbidities in WAG/Rij rats has been increasingly acknowledged. A. Leo et al. (2019) concluded that cognitive impairment and depressive-like behavior in WAG/Rij rats waere secondary to the occurrence of absence seizures [77]. These authors also suggested that absence seizures are necessary for the expression of cognitive impairment.

Cognitive comorbidities of absence epilepsy in WAG/Rij rats included cognitive impairment, depression-like symptoms, and altered emotional responsiveness (Table 1). K. Sarkisova et al.’s comprehensive studies performed in 2003–2011 [76,96,97] indicated that WAG/Rij rats display a range of behavioral signs of comorbid depression, as well as being particularly sensitive to stress, making them an ideal candidate for modeling chronic low-grade depression. An in-depth analysis of A. Leo et al. (2019) [77] indicated the following:Cognitive impairment in WAG/Rij rats was secondary to absence epilepsy and to depressive-like behavior;Absence epilepsy, depressive-like behavior, and cognitive deficit may arise independently and separately in a lifetime from the same underlying network disease;Cognitive impairment in WAG/Rij rats was age-dependent and was linked to the age-dependent increase in spike-wave discharges (i.e., the electroencephalographic sign of absence epilepsy).

Fedosova at al. (2015) [98] found that WAG/Rij rats display increased anxiety and higher stress responses compared to Wistar rats, which precedes the emergence of absence epilepsy, depression-like behavior, and cognitive impairments.

Midzyanovskaya et al. (2005) tested WAG/Rij rats in the Light–dark Choice test and the Open Field test [78]. The Light–dark Choice test, in contrast to the Open Field test, allows an option to escape to a dark compartment. Therefore, the Light–dark Choice test is considered less stressful for rats than the Open Field test. In the Light–dark Choice test, WAG/Rij rats showed an increased locomotor activity (i.e., more entries into the light compartment and more rearings) and more emotional responses (i.e., defecation and urination) than Wistar rats. In contrast, in the Open Field test, WAG/Rij rats were more passive than Wistar rats, and showed reduced exploration and more episodes of grooming [78]. The authors stated that *“mild environmental stressors can stimulate the hyperresponsive nucleus accumbens of absence-epileptic rats and produce exaggerated behavioral response”* [78]. In the WAG/Rij rats, an exaggerated behavioral response to a novel environment was observed, which somewhat contradicted the findings of depressive-like behavior. Increased grooming in the Open Field in WAG/Rij rats might indicate an increased level of anxiety.

On the contrary, Karson et al. (2012) showed no significant differences between WAG/Rij and Wistar rats at the ages of 5 months and 13 months in terms of anxiety-like behavior and locomotor activity [79]. However, the emotional and spatial memory in WAG/Rij rats at the age of 13 months was poorer than in age-matched Wistar rats as determined using passive avoidance and Morris water maze tests [79]. In addition, WAG/Rij rats showed a poorer retention performance in the passive avoidance test than Wistar rats at the ages of 2 and 6 months [81].

Less than 100% of WAG/Rij rats showing anxiety, but the exact percentage has not been specified. Kliueva et al. (1999) found that the anxiety in WAG/Rij rats varied depending on their susceptibility to audiogenic seizures [99]. However, the studies of Sarkisova et al. [100] reported that only the rats which were susceptible to audiogenic seizures showed an increase in anxiety, while the non-susceptible rats did not.

We investigated the dynamics of fear-motivated associated learning in an Active Avoidance test using epileptic and non-epileptic WAG/Rij rats at the age of 6 months [83]. Our data indicated that there was a slight deficit in executive functions during associative learning in adulthood. In rats with severe absence epilepsy, a higher time spent in SWDs positively correlated with poorer test performance, suggesting a deficit in short-term memory in subjects with severe epilepsy. WAG/Rij rats with a depressive phenotype may have deficient executive function. We recently conducted an in-depth analysis of associative learning in parallel with the recording of ultrasonic vocalization in epileptic WAG/Rij rats and NEW rats (Non-Epileptic WAG/Rij, a minor substrain selected from WAG/Rij) [82]. Both strains had a genetic predisposition to absence epilepsy, but the WAG/Rij rats exhibited seizures and the minor NEW substrain were non-epileptic. The strain difference in Active Avoidance learning was found only between males, not between females. WAG/Rij male rats demonstrated a poorer test performance than non-epileptic NEW males. The aversive ultrasonic vocalization (22–25 kHz) did not differ between strains (only sex differences were reported). Freezing was found in one exceptional female WAG/Rij rat (3.7% of females) and in 40% of the males of both strains (these rats were not able to perform the task). Perhaps, WAG/Rij rats and a minor NEW substrain show an increased propensity for emotional disorders or emotional exhaustion, and, therefore, they are unusually sensitive to emotional stimuli.

A recent meta-analysis of psychiatric and neurological comorbidities in patients with absence epilepsy showed a bidirectional relationship between depression and anxiety [80]: “*patients with depression and anxiety are at increased likelihood of a diagnosis of epilepsy later in life, and epilepsy is associated with a higher incidence of various psychiatric disorders such as depression, anxiety, and suicidal ideation*”.

**Table 1 biomedicines-12-00122-t001:** Learning and behavior (cognitive comorbidities) in WAG/Rij rat model of absence epilepsy.

Subjects	Tests	Test Results	References
Male WAG/Rij rats; Untreated and ethosximide-treated (300 mg/kg/day; 17 days); 6 and 12 months old	The Forced Swimming test The novel objects recognition test Social recognition test Morris water maze Passive Avoidance	Anxiety, learning and behavior (cognitive skills) The Forced Swimming test. WAG/Rij rats exhibited an increased immobility time. The novel objects recognition test for short-term (working) memory. WAG/Rij rats had a lower discrimination index than Wistar rats, indicating a working memory deficit. Additionally, WAG/Rij rats had a lower total distance moved, suggesting a locomotor deficit. Social recognition test for recognition memory (familiar vs novel juvenile male rat). WAG/Rij rats showed lower social recognition indices compared to Wistar rats, indicating deficit of short- and long-term recognition memory. Morris water maze for learning and memory. No differences at the age of 6 months. WAG/Rij rats at the age of 12 months showed longer escape latency and less time in the target quadrant in than Wistar rats, suggesting an impairment of spatial reference memory in 12-m old WAG/Rij rats. Passive avoidance—a fear-motivated test for long-term memory. No differences at the age of 6 months. WAG/Rij rats at the age of 12 months showed a longer retention (24 h) of passive avoidance response, suggesting deficit of long-term memory.	Leo et al., 2019 [77]
Drug-naive adult male WAG/Rij rats and Wistar control rats, data compilation	The Open Field test The Forced Swimming test Sucrose Preference test for anhedonia (20% solution, two-bottles choice) The Light–Dark choice test The social interaction test in the open field The elevated plus-maze.	Anxiety and behavior The open field. WAG/Rij rats showed lower explorative activity and lower number of groomings than Wistar rats. The Forced Swimming test. WAG/Rij rats showed longer immobility, shorter first swimming episode and shorter duration of total swimming. Sucrose (20%) Preference test. The preference for sucrose in WAG/Rij rats (65.1%) was lower than in Wistar rats (91.9%). TheLight–Dark choice test. No differences between WAG/Rij and Wistar rats, except one—the number of rearings in the light compartment of the light–dark box during the 2nd test session in WAG/Rij rats was lower than in Wistar rats. The social interaction test. Duration of active social contacts in WAG’Rij rats was lower than in Wistar rats. The elevated plus-maze. No differences between WAG/Rij and Wistar rats.	Sarkisova and van Luijtelaar, 2011 [76] Sarkisova et al., 2003 [97]
Drug-naive male WAG/Rij rats and Brown Norway control rats; 13 months old	Test for spatial memory in 16-holes field (holeboard)	Spatial memory (working memory and reference memory The working memory in WAG/Rij rats was better than in Brown Norway rats. The reference memory in WAG/Rij rats was poorer than in Brown Norway rats.	van der Staay, 1999 [101]
Drug-naïve WAG/Rij rats, outbred and Wistar rats; adult and 2 months old	The Elevated Plus Maze	Anxiety and behavior The Elevated Plus Maze. WAG/Rij rats demonstrated fewer visits to the open arms, less rearing activity, and more episodes of freezing than Wistar and outbred rats.	Kliueva et al., 1999 [99]
Drug-naive female WAG/Rij rats and Wistar control rats; approx. 4 months old	The Open Field test of mothers and their pups on the postnatal days 4–6	Maternal behavior The Open Field test. WAG/Rij rat mothers less frequently approached their pups and moved slower than Wistar mothers.	Dobryakova et al., 2008 [102]
Drug-naive male WAG/Rij rats and Wistar control rats; 2 months old	The Open Field test The Forced Swimming test The Light–dark choice test Two-days passive avoidance learning test Two-days Active Avoidance learning test	Behavior and learning on the preclinical stage. The Open Field test. No differences between WAG/Rij and Wistar rats. The Forced Swimming test. No difference in duration of swimming, in the number of dives, head twitches and the immobility time between WAG/Rij and Wistar rats. The duration of the first episode of active swimming in WAG/Rij rats was shorter than in Wistar rats. The Light–dark choice test. No difference between WAG/Rij and Wistar rats in 5 out of 6 measures. Only one score in WAG/Rij rats was higher than in Wistar rats: the number of unsuccessful attempts to enter the light compartment (so-called “risk assessments”). The Passive Avoidance test. No difference between WAG/Rij and Wistar rats. The Active Avoidance test. On the 1st day, WAG/Rij rats showed more avoidances than Wistar rats. On the 2nd day, WAG/Rij and Wistar rats showed the same number of avoidances. The memory trace storage in WAG/Rij rats was -8.26%, and in Wistar rats it was 283.33%.	Fedosova et al., 2015 [98]
Drug-naive male WAG/Rij rats with epileptic and non-epileptic phenotypes; 6 months old	The Active Avoidance test	The Active Avoidance test on the clinical stage (fear-motivated associative learning). In contrast to non-epileptic phenotype, epileptic WAG/Rij rats showed deficit of executive functions, rather than impairment of memory. A higher time spent in SWDs positively correlated with poorer test performance, suggesting a deficit in short-term memory in subjects with severe epilepsy, but not in rats with mild epilepsy.	Sitnikova & Smirnov, 2020 [83]
Drug-naive male and female WAG/Rij rats and control non-epileptic NEW rat substrain; 8.53 ± 1.15 months old	The Active Avoidance test	The Active Avoidance test on the clinical stage (fear-motivated associative learning). Both strains were prone to absence epilepsy, with WAG/Rij rats exhibiting seizures and a minor NEW substrain being non-epileptic. In the WAG/Rij male rats, avoidance learning was significantly poorer than in non-epileptic NEW males, but no strain difference was found between females. Freezing in one exceptional female WAG/Rij rat (3.7% of females) and in twelve males of both strains (40% of males); these rats were not able to perform the task.	Alexandrov et al., 2023 [82]

## 5. Behavioral and Cognitive Comorbidities in the GAERS

As well as WAG/Rij rats, the GAERS also share some similarities with human patients in terms of anxiety and depressive-like behavior. Some of the most significant findings, observations, and conclusions are noted below.

Jones et al. (2008) examined behavioral comorbidities of absence epilepsy in the GAERS [85] and reported that the GAERS differed from NECs by the following:Reduced consumption of 20% sucrose solution;Spending less time in the open arms of the Elevated Plus Maze;Reduced exploratory activity in the Open Field test;Spending less time in the inner area of the Open Field test.

In summary, all measures presented in this study revealed significantly greater levels of both depression- and anxiety-like behaviors in the GAERS [85].

Marques-Carneiro et al. (2014) compared the locomotion and anxiety in the GAERS at the age of 3–6 months, in the NECs as the first control group, and in Wistar rats as the second control group [72]. This study indicated the following:All three strains showed similar levels of locomotor activity as measured in their home cages during the lights-on period;The NECs and the GAERS were slightly less active in their home cages than Wistar rats during the light-off period;In the beam-walking test, the GAERS and the NECs showed good sensorimotor abilities. Among the three strains, Wistar rats showed the poorest sensorimotor abilities, likely because the body weight in Wistar rats exceeded that in the GAERS and the NECs;The GAERS showed a higher anxiety than NECs in the Open Field test (lower activity scores in both central and peripheral areas, and a lower number of rearings) and in the Plus Maze test (a lower number of entries in open arms). However, the results of the GAERS did not differ from those of the Wistar rats;When exposed to higher novelty in the Open Field, the GAERS showed a reduced exploration, as compared to the NECs and Wistar rats.

Marks et al. (2016) investigated anxiety, sensorimotor gating, and cognitive performance in male and female GAERS and NECs during the prepubescent age (P35) and young adult age (P56) [103]. The GAERS spent less time in the open arms of the Elevated Plus Maze and showed an elevated startle response, and this was interpreted as an anxiety-like behavior. Furthermore, an increased footshock reactivity in the GAERS rats as well as enhanced freezing to conditioned fear-associated cues confirmed anxiety-like responses. The GAERS showed the following differences from the NECs:During prepuberty, both sexes spent less time in open arms and had fewer total open and closed arm entries in the Elevated Plus Maze;During prepuberty and young adulthood, both sexes travelled less distance in both the inner and outer areas of the Open Field;During young adulthood, females spent less time in open arms in the Elevated Plus Maze, with no difference between the males;During prepuberty and young adulthood, both sexes exhibited higher startle responses;During prepuberty and young adulthood, males showed increased freezing relative in the low-intensity fear conditioning;Exaggerated cued and contextual Pavlovian fear conditioning and impaired fear extinction;An impairment of latent inhibition in a paradigm using Pavlovian fear conditioning.

In summary, this study [103] demonstrated increased anxiety-like behavior, altered exploration in the open field, increased acoustic startle and increased prepulse inhibition, increased freezing during conditioning; and impaired extinction of conditioned fear in the GAERS.

Recently, Neuparth-Sottomayor et al. (2023) published a comprehensive report [104] in which they described results of test battery for anxiety, and short- and long-term memory. They used GAERS and two control groups—Wistar rats and non-epileptic control rats (<30 subjects per group at the age of 3–6 months). The GAERS exhibited the following neuro-behavioral peculiarities.

Deficits in working, spatial reference, and recognition memory as compared to both NEC and Wistar rats;Did not show an exaggerated anxiety-like phenotype, but rather a lower anxiety-like behavior in two out of three anxiety tests;Preferentially used egocentric strategies to perform spatial memory tasks.

## 6. The Thalamocortical Network and “the Cognitive Thalamus”

As was already mentioned, the thalamocortical neuronal circuitry has long been known as the primary source of absence seizures [10,11,13,14,15]. Recent studies have identified fine changes in cortical and thalamic activity that underlie absence epilepsy, as reviewed by [61,105,106].

The thalamus is an important sensory gateway of the brain, controlling the flow of sensory-motor information to the cerebral cortex [10,17,23,107]. The thalamus acts as a relay station and gatekeeper, and its dysfunction is associated with various neuropsychological and brain disorders [108]. The thalamus is globally connected with distributed cortical regions and is capable of integrating multimodal information across diverse cortical functional networks [109]. In their review, M. Wolff and S.D. Vann (2019) clearly demonstrated that the thalamus plays a complex role in cognition, including learning, memory, and flexible adaptation [19]. These authors used the term “*the cognitive thalamus” suggesting the role of higher-order and limbic thalamic nuclei in cognition: “from a behavioral perspective, the term “cognitive thalamus” more accurately captures the essence of those thalamic nuclei that primarily support cognitive functions*” [19]. In general, the thalamus is involved in multiple cognitive functions and is a critical hub region that could integrate diverse information being processed throughout the cerebral cortex as well as maintain the modular structure of cortical functional networks. R. Llinás et al., in 1999, introduced the concept of thalamocortical dysrhythmia: a neurological and neuropsychiatric syndrome characterized by a resonant interaction between the thalamus and cortex, resulting in coherent theta activity due to the generation of low-threshold calcium spike bursts by thalamic cells [27]. It is important to note that thalamocortical dysrhythmia does not simply mean that the thalamus is dysfunctional, but that the cortex and thalamus are dysregulated together. This mechanism has been proposed to underlie diverse neurological disorders such as Parkinson’s disease, neuropathic pain, tinnitus, and depression [110,111,112].

In short, epileptic activity in thalamocortical and corticothalamic networks can lead to significant disruptions in normal cognitive functions, such as attention, memory, and executive functions [34,36,105].

## 7. Some Translational Issues

The application of the translational approach has become more and more common due to its potential to link animal studies with clinical practice, and to incorporate the latest breakthroughs in human medicine. Rats are nocturnal animals. They are more awake during the dark period and less active during the light period. Despite this, behavioral studies are usually done during the light period, in a lighted environment. This might not be a crucial issue, because rats have a polyphasic sleep pattern, which means it is natural for them to be awake during the light phase of the day. An important issue is the endogenous circadian mechanism, which controls the timing of spontaneous SWDs in rats [42,113,114]. SWDs in WAG/Rij rats were most frequent in the middle of the dark period [114]. Behavioral experiments are usually conducted during the light phase, when the number of SWDs varies between medium and high [113,114]. Recently, C. McCafferty et al. [75] described prolonged brain state changes preceding absence seizures, namely changes in neuronal firing in the cortex and the thalamus, occurring 40–60 s before SWD onset. Therefore, the brain’s functioning in epileptic subjects is impaired, resulting in significant behavioral consequences.

Attention deficits have been recognized as the most common comorbidity, affecting an estimated 35–40% of patients with absence epilepsy [105,115]. Most seizure-free patients experienced some cognitive difficulties after successful pharmacological treatment with ethosuximide, valproic acid, and lamotrigine [44,45]. Ethosuximide is a first-choice anti-absence drug that also prevented seizures in WAG/Rij rats and improved cognitive and behavioral functions in these rats to healthy levels [68,96]. However, the positive effect of ethosuximide was only temporary, and absence seizures with associated behavioral comorbidities returned after 5 months of drug suspension [116]. Eventually, the pharmacological treatment did not cure the genes related to absence epilepsy and comorbidities persisted.

Next to difficulties in executive function (more generally, attention) [83], poorer associative learning in WAG/Rij rats [79,81,82] might be accounted for by a lack of motivation to explore the environment resulting from depressive-like symptoms [68,76,80,96]. It is an open question for future research whether the slower acquisition of learning seen in WAG/Rij rats is primarily caused by a deficit in executive function or by a motivational impairment.

Another issue is a putatively high level of emotionality in WAG/Rij rats, which makes them more likely to have negative experiences. WAG/Rij rats demonstrated behavioral signs of emotional excitation in the Ligh-Dark choice test, such as an increased locomotion and defecation/urination [78]. “*It is likely that genetic absence epilepsy, especially when a mixed pathology is present, is accompanied by a high vulnerability to stressors*” [78]. Our analysis led us to the same assumption: genetic predisposition to absence epilepsy links to a propensity for emotional disorders or emotional exhaustion [82]. It is possible that WAG/Rij rats were unusually sensitive to novel emotional stimuli, due to their unique genetic background.

The next challenging task is defining measurable predictors of cognitive problems in absence epilepsy or biomarkers of cognitive comorbidities. In order to find biomarkers of thalamocortical disfunction in relation to cognitive functioning, we examined correlations between intrinsic characteristics of SWDs and parameters of associative learning [83]. Six-month-old male WAG/Rij rats with different severities of absence epilepsy were used, and it was concluded that the intrinsic frequency of SWDs could not be used as a marker of cognitive dysfunction in absence epilepsy. A lower intrinsic frequency of SWDs (in proportion to a reduction of instantaneous frequency at the end of seizures) may be considered as a sign of attention problems in epileptic rats.

## 8. Conclusions

This review focuses on natural/genetic models of absence epilepsy, in which pathological processes are genetically predetermined, such as WAG/Rij rats [32,38] and GAERS [31,47,103]. Patients with absence epilepsy are typically treated with pharmacotherapy; therefore, studies in drug-naive rats with spontaneous absence epilepsy offer a valuable opportunity to better understand brain dysfunctions. Rat models exhibited *behavioral comorbidities* which are expressed as depression- and anxiety-like symptoms and *cognitive comorbidities*—poor learning abilities. Here, I overview results obtained in the battery of behavioral tests, including the Forced Swimming test, the Tail Suspension test, the Sucrose Preference test, the Light–Dark choice test, the Elevated Plus Maze test, the Open Field test, the novel objects recognition, social recognition, and social interaction test, and various tests for memory and learning skills.

Most of the studies in the GAERS demonstrated depression- and anxiety-like symptoms. It is generally believed that WAG/Rij rats also have depression-like symptoms, but there are some doubts regarding anxiety-like behaviors in WAG/Rij rats.

Caution should be taken when using control rats as a non-epileptic control for epileptic strains. Some rat strains, such as Wistar, Sprague–Dawley, and Long Evans, exhibit spontaneous absence seizures in adulthood [70,73,117,118,119].

Absence seizures are caused by the impairment of the thalamocortical neuronal circuitry; therefore, the concept of the “cognitive thalamus” could explain some cognitive comorbidities. The thalamus is involved in multiple cognitive functions, and epileptic activity in the thalamocortical circuitry can disrupt normal cognitive functions, such as attention, memory, and executive functions.

This review discusses current challenges in the field, such as putatively high levels of emotionality in genetically prone rats, sex differences in their expression of cognitive comorbidities, and predictors of cognitive problems or biomarkers of cognitive comorbidities in absence epilepsy. Data obtained in genetic rat models of absence epilepsy improved our understanding of cognitive dysfunction associated with this disease.

## Figures and Tables

**Figure 1 biomedicines-12-00122-f001:**
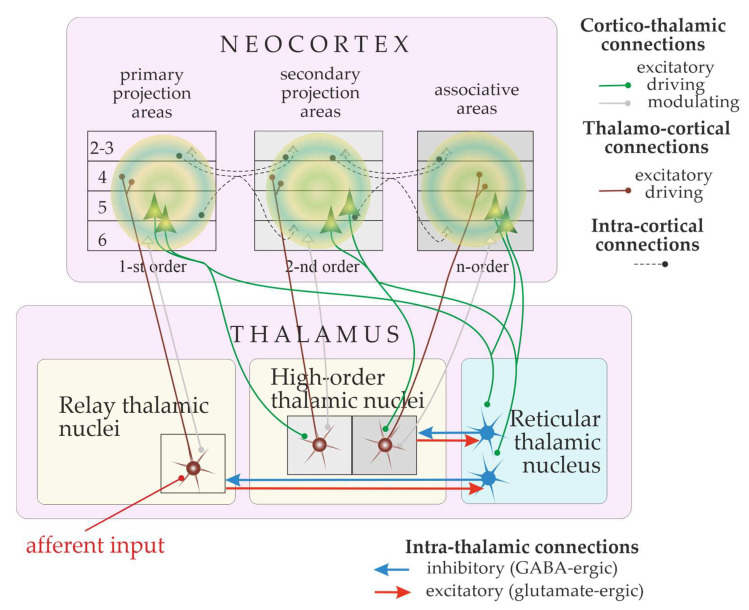
Schematic representation of the thalamocortical system. The loop is activated by external stimulus (afferent input). Thalamo-cortical ascending connections are shown in brown. Cortico-thalamic descending connections are shown in green (driving inputs) and grey (modulatory inputs). The terminology is from [16].

**Figure 2 biomedicines-12-00122-f002:**
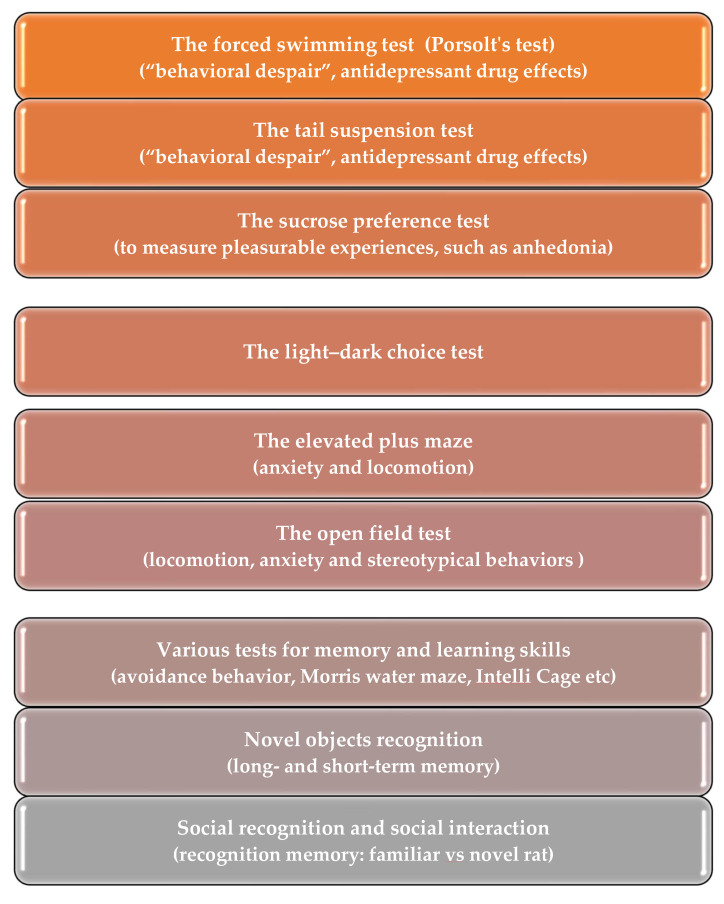
Behavioral test battery used to assess behavioral comorbidities in genetic rat model of absence epilepsy.

## Data Availability

Not applicable.

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
