# Peer review of "Behavioral and Cognitive Comorbidities in Genetic Rat Models of Absence Epilepsy (Focusing on GAERS and WAG/Rij Rats)"

_biomedicines, 2024, doi:10.3390/biomedicines12010122_

Round 1
Reviewer 1 Report
Comments and Suggestions for Authors
I would regard this paper as providing a fairly extensive review, but not a critical review, of behavioural and cognitive alterations in two species of genetically absence seizure prone rats. It is generally well organised and the English not difficult to follow, though there are occasional repetitions and curious spellings that could be dealt with in house. The paper brings together a collection of material from various sources that would be helpful to workers in the area and to other readers, particularly the account of the various animal models used in studies in the area and how these are interpreted. Unfortunately the paper provides no numerical data at all, and no results of statistical significance testing though the latter may not be a conventional requirement in the paper’s research area.
On balance, the above would probably suffice to commend the paper for publication but to my mind a crucial issue remains, viz. how can one be sure that the different behaviours in the seizure-prone animals as compared with controls are not simply due to their being assessed during actual or subclinical seizures. Children in absence status can behave (unnaturally) like little angels and the reason be detected only electrographically. I think the author needs to deal with that point in some way or other before a decision is taken regarding acceptance, while numerical results would add to the paper’s value.
I see a potentially critical issue in the interpretation of the findings of the investigations cited. How can it be excluded that the alterations found are not due to undetected absence seizures occurring while the observations are being made rather than consequences of a seizure disorder that was inactive at the times of the observations? For instance, was brain electrical activity being recorded simultaneously? In childhood absence status children can behave unnaturally well, and the basis of this altered behaviour go unrecognised without EEG recording.
In the paper the term ‘spike-wave’ is used repeatedly but it is never indicated where in the head/brain the appearance occurs. Local spike-wave appearances occasionally occur in human focal epilepsies.
You sometimes write of ‘absences’ and sometimes of ‘epilepsy’. I assume you are using different words for the same phenomena but humans with absence seizures may also have generalised convulsive ones and I wonder if it might be wiser to write of ‘absence epilepsy’ if that is what you intend.
Comments on the Quality of English Language
There is a little repetition at one stage, and a mis-spelling,
Author Response
Dear Editor,
I am deeply grateful for your decision on my manuscript, as well as the feedback and comments. Thank you for your prompt and thorough suggestions, which I have taken into consideration for this revision (highlighted in the MS). Please find my response below.
The paper brings together a collection of material from various sources that would be helpful to workers in the area and to other readers, particularly the account of the various animal models used in studies in the area and how these are interpreted. Unfortunately the paper provides no numerical data at all, and no results of statistical significance testing though the latter may not be a conventional requirement in the paper’s research area.
On balance, the above would probably suffice to commend the paper for publication but to my mind a crucial issue remains, viz. how can one be sure that the different behaviours in the seizure-prone animals as compared with controls are not simply due to their being assessed during actual or subclinical seizures. Children in absence status can behave (unnaturally) like little angels and the reason be detected only electrographically. I think the author needs to deal with that point in some way or other before a decision is taken regarding acceptance, while numerical results would add to the paper’s value.
Response. I agree that providing precise numerical results would make my review more useful and accurate. I have no idea what numerical data I could add (I mentioned precise ages, percentages, doses etc, especially data obtained by my group). Meanwhile, I added on page 6 the paragraph with numerical data about anxiety: “Less than 100% WAG/Rij rats showing anxiety...”.
I see a potentially critical issue in the interpretation of the findings of the investigations cited. How can it be excluded that the alterations found are not due to undetected absence seizures occurring while the observations are being made rather than consequences of a seizure disorder that was inactive at the times of the observations? For instance, was brain electrical activity being recorded simultaneously? In childhood absence status children can behave unnaturally well, and the basis of this altered behaviour go unrecognised without EEG recording.
Response. This is true for CAE, but it is not true in rats. Let me answer this question using the citation from a recent paper published in the highly impactful journal Neurobiology of Disease in 2023 (10.1016/j.nbd.2023.106275). The authors used GEARS and noted: “One potential limitation of behavioural studies in epileptic animals is that the occurrence of seizures during the tests could markedly affect the animal's performance. However, it is well established that absence seizures do not occur when an animal is involved in active exploration but are present almost exclusively when the animal is in a behavioural state of quite wakefulness (Coenen et al., 1991; Danober et al., 1998).”
In order to clarify this issue, I added to the text (page 2):
“Earlier studies in genetically prone rats found almost no SWDs during cognitive task [doi:10.1016/0013-4694(91)90208-L]. Only 4% of SWDs appear during active behavior [doi:10.1016/0920-1211(91)90055-K]. Moreover, genetically prone rats were able to discriminate meaningful and meaningless stimuli presented during SWDs [39]. Therefore, cognitive comorbidities could be considered consequences of a seizure disorder. “
For instance, was brain electrical activity being recorded simultaneously?
Response. In order to clarify this issue, I added to the text (page 2): “Spontaneous absence seizures in genetic rat models could only be detected using electroencephalogram (EEG) by the presence of high-voltage spike-wave discharges (SWDs) [doi:10.3389/fphar.2020.00395; Depaulis & van Luijtelaar, 2006; doi:10.1111/j.1528-1167.2007.01250.x]. Monitoring of EEG in freely moving rats requires chronically implanted electrodes, which is rather invasive.”
In the paper the term ‘spike-wave’ is used repeatedly but it is never indicated where in the head/brain the appearance occurs. Local spike-wave appearances occasionally occur in human focal epilepsies.
Response. I tried to use less acronyms and used "spike-wave seizures" as a synonym for SWDs. Now I fixed the text for the term "spike-wave".
You sometimes write of ‘absences’ and sometimes of ‘epilepsy’. I assume you are using different words for the same phenomena but humans with absence seizures may also have generalised convulsive ones and I wonder if it might be wiser to write of ‘absence epilepsy’ if that is what you intend.
Response. Thank you for the clarification. I used the term “absence seizure(s)” as a synonym for EEG/behavioral absence (or absence epileptic attack). I understand the problem and used the term “absence epilepsy” wherever appropriate.
There is a little repetition at one stage, and a mis-spelling
Response. I proofread the paper, found and corrected some errors.
Reviewer 2 Report
Comments and Suggestions for Authors
Absence epilepsy is an important epileptic syndrome in children, however, the mechanism is unknown. Animal study in drug-naive rats with spontaneous absence epilepsy offer a valuable opportunity to better understand brain dysfunctions. Absence epilepsy is less common than convulsive type of epilepsy, but it can be just as dangerous if not treated. In view the fact that absence seizures could occur without warning, they could be extremely frightening for patients and their families. Consequently, there is a considerable concern regarding the link between epilepsy and a range of neuropsychiatric comorbidities, such as anxiety, depression, and attention-deficit/hyperactivity disorder.
This review focuses on natural/genetic models of absence epilepsy, in which pathological processes are genetically predetermined, such as WAG/Rij rats and GAERS. Most of the studies in the GAERS demonstrated depression- and anxiety-like 438 symptoms. It is generally believed that WAG/Rij rats also have depression-like symp- 439 toms, but there are some doubts on anxiety-like behaviors in WAG/Rij rats.
In addition, the review discusses current challenges in the field, such as putatively high level of emotionality in genetically prone rats, sex differences in expression of cognitive comorbidities, predictors of cognitive problems or biomarkers of cognitive comorbidities in absence epilepsy.
In conclusion, I think it is valuable for our readers.
Comments on the Quality of English Language
In conclusion, I think it is valuable for our readers.
Author Response
Dear Reviewer,
Thank you for your positive feedback and for the very high evaluation of my review. I really appreciate your kind words and am grateful for your support.
I am to your opinion regarding the focus of my review on natural/genetic models of absence epilepsy, in which pathological processes are genetically predetermined, such as WAG/Rij rats and GAERS. As well as to the challenging problem in the field, such as putatively high level of emotionality in genetically prone rats.
Reviewer 3 Report
Comments and Suggestions for Authors
Presented manuscript provides a comprehensive review of the neurocognitive comorbidities in 2 specific genetic rat models of absence epilepsy. The review focuses on on cognitive and behavioral comorbidities in two generally accepted genetic rat models of absence epilepsy: the inbred Wistar Albino Glaxo Rats from Rijswijk (WAG/Rij) and in the Genetic Absence Epilepsy Rats from Strasbourg (GAERS). Since review is only addressing 2 genetic models it would be advisable to specify that in the title. The manuscript is well organized and contains clear and concise review of important scientific information related to presented subject.
Recently published study (Mariana Neuparth-Sottomayor, Carolina C. Pina, Tatiana P. Morais, Miguel Farinha-Ferreira, Daniela Sofia Abreu, Filipa Solano, Francisco Mouro, Mark Good, Ana Maria Sebastião, Giuseppe Di Giovanni, Vincenzo Crunelli, Sandra H. Vaz. Cognitive comorbidities of experimental absence seizures are independent of anxiety. Neurobiology of Disease, Volume 186, 2023, 106275, ISSN 0969-9961) contains new evidence of memory deficits in GAERS rats that do not depend on an anxiety or neophobic phenotype. In addition, authors of this study, contrary to some previous observations, suggest that GAERS rats do not exhibit anxiety-like behavior and neophobia compared to both NEC and Wistar rats. I suggest inclusion of this recently published observation in presented manuscript.
Minor concerns:
1. Verse 12 correct Strasburg (GAERS) to Strasbourg (GAERS)
Author should check all sentences where last names are included and remove capital letters that sand for the first name initials of cited authors. Some examples below:
Verse 52 Remove “H” before Blumenfeld
Verse 97, 99 and 102 “P” before cited last name should be removed.
Verse 109 letters “G” and “A” before cited last names should be removed.
Verse 123, 127, 129 remove “J” before cited last names.
2. Verse 124 remove “(“.
3. Verse 212 has a part of the word “outcome” italicized.
4. In abbreviations the terms: NEW, NEC were not included.
Author Response
Dear Editor,
I am deeply grateful for your decision on my manuscript, as well as for the feedback and comments. Please find my response below.
Presented manuscript provides a comprehensive review of the neurocognitive comorbidities in 2 specific genetic rat models of absence epilepsy. The review focuses on on cognitive and behavioral comorbidities in two generally accepted genetic rat models of absence epilepsy: the inbred Wistar Albino Glaxo Rats from Rijswijk (WAG/Rij) and in the Genetic Absence Epilepsy Rats from Strasbourg (GAERS). Since review is only addressing 2 genetic models it would be advisable to specify that in the title.
Response. Yes, indeed. My review focuses on WAG/Rij and GAERS. Accordingly, I changed the title “Behavioral and cognitive comorbidities in genetic rat models of absence epilepsy (focusing on GAERS and WAG/Rij rats)”
The manuscript is well organized and contains clear and concise review of important scientific information related to presented subject.
Recently published study (Mariana Neuparth-Sottomayor, Carolina C. Pina, Tatiana P. Morais, Miguel Farinha-Ferreira, Daniela Sofia Abreu, Filipa Solano, Francisco Mouro, Mark Good, Ana Maria Sebastião, Giuseppe Di Giovanni, Vincenzo Crunelli, Sandra H. Vaz. Cognitive comorbidities of experimental absence seizures are independent of anxiety. Neurobiology of Disease, Volume 186, 2023, 106275, ISSN 0969-9961) contains new evidence of memory deficits in GAERS rats that do not depend on an anxiety or neophobic phenotype. In addition, authors of this study, contrary to some previous observations, suggest that GAERS rats do not exhibit anxiety-like behavior and neophobia compared to both NEC and Wistar rats. I suggest inclusion of this recently published observation in presented manuscript.
Response. Thank you for this suggestion. I added this recent paper to my review. On page 11:
“Recently (2023) Neuparth-Sottomayor et al published a comprehensive report [100], in which they described results of test battery for anxiety, short- and long-term memory. They used GAERS and two control groups - Wistar rats and non-epileptic control NEC rats (< 30 subjects per group at the age of 3–6 months). The GAERS exhibited the following neuro-behavioral peculiarities.
- Deficits in working, spatial reference and recognition memory as compared to both NEC and Wistar rats.
- Did not show an exaggerated anxiety-like phenotype, but rather a lower anxiety-like behavior in two out of 3 anxiety tests.
- Preferentially used egocentric strategies to perform spatial memory tasks. “
Minor concerns:
- Verse 12 correct Strasburg (GAERS) to Strasbourg (GAERS)
Response. Thank you. It is corrected.
Author should check all sentences where last names are included and remove capital letters that sand for the first name initials of cited authors. Some examples below:
Response. Corrected.
Verse 52 Remove “H” before Blumenfeld
Verse 97, 99 and 102 “P” before cited last name should be removed.
Verse 109 letters “G” and “A” before cited last names should be removed.
Verse 123, 127, 129 remove “J” before cited last names.
- Verse 124 remove “(“.
- Verse 212 has a part of the word “outcome” italicized.
Response. Corrected.
- In abbreviations the terms: NEW, NEC were not included.
Response. Thank you for mentioning this problem. It is corrected now. I added NEW and NEC to abbreviation list.
Reviewer 4 Report
Comments and Suggestions for Authors
The manuscript “Behavioral and cognitive comorbidities in genetic rat models of absence epilepsy”, by Evgenia Sitnikova, review the cognitive features in two strains of rats with genetic absence epilepsy, i.e, GAERS and WAG/Rij. The author affirms, “the knowledge of behavioral and cognitive comorbidities in drug-naive rats with spontaneous absence epilepsy is beneficial for understanding the pathophysiology of absence epilepsy, and for finding new treatment strategies”. Probably all the researchers in the field would agree with this sentence but, in this manuscript the link between comorbidities and pathophysiology is poorly present.
Introduction is interesting and well written.
The description of behavioural and cognitive features of both strain of rats are informative, but I find a poor systematization. Probably, it would be informative for majority of readers unfamiliar with translational research a more straight comparison between both strains and not only enlisted of features.
Considering that the presence or absence of epilepsy is the most relevant feature of the state of rats, I think that it would be appropriate to briefly describe the method to identify the presence of epilepsy. Obviously the performance of rats in the test would be different whether they suffer or not a seizure. Therefore, a brief description of method to record EEG should be detailed.
Are or not the NEW rats strain epileptic?, because in line 273 it’s stated that both strains were prone to absence epilepsy. This point needs to be clarified and explain the role of this strain.
The description of the thalamocortical network, in fact, is mainly based in Introduction, with the same references and, therefore in a high degree can be considered as reiteration, because scarcely new information is offered. However, I think that a good description of the thalamocortical system, including a diagram, would be extremely important. A detailed information about how the malfunction of this network can explain the features of rats would be really important and useful for readers, mainly clinicians.
Some minor comments:
1.- Line 36. Absence seizures occur without warning but the rest of seizures occur in the same way. Please, modify this sentence.
2.- Please, indicate the references supporting the lines 58-60, because, in fact, this is a very relevant aspect of the manuscript.
3.- Lines 102-103. Please, clarify the concepts of face and predictive validities.
4.- Line 296. Define the acronym NEC.
Author Response
Dear Editor,
I am deeply grateful for your decision on my manuscript, as well as for the feedback and comments. Thank you for your prompt and thorough suggestions, which I have taken into consideration for this revision. Please find my response below.
The description of behavioural and cognitive features of both strain of rats are informative, but I find a poor systematization. Probably, it would be informative for majority of readers unfamiliar with translational research a more straight comparison between both strains and not only enlisted of features.
Response. There are some subtle differences between WAG/Rij rats and GEARS, but these are some competition between these strains. In order to avoid any potential conflict of interest, I will not make a direct comparison between the two strains.
Considering that the presence or absence of epilepsy is the most relevant feature of the state of rats, I think that it would be appropriate to briefly describe the method to identify the presence of epilepsy. Obviously the performance of rats in the test would be different whether they suffer or not a seizure. Therefore, a brief description of method to record EEG should be detailed.
Response. In order to clarify this issue, I added one paragraph to Introduction (Page 2): “Spontaneous absence seizures in genetic rat models could only be detected using electroencephalogram (EEG) by the presence of high-voltage spike-wave discharges (SWDs) [34–36]. Monitoring of EEG in freely moving rats requires chronically implanted electrodes, which is rather invasive.”
Are or not the NEW rats strain epileptic?, because in line 273 it’s stated that both strains were prone to absence epilepsy. This point needs to be clarified and explain the role of this strain.
Response. Thank you for noting this uncertainty. I corrected the phrase on Page 7: “Both strains were prone had genetic predisposition to absence epilepsy, with but WAG/Rij rats exhibiting exhibite seizures and a minor NEW substrain being non-epileptic.”
The description of the thalamocortical network, in fact, is mainly based in Introduction, with the same references and, therefore in a high degree can be considered as reiteration, because scarcely new information is offered. However, I think that a good description of the thalamocortical system, including a diagram, would be extremely important. A detailed information about how the malfunction of this network can explain the features of rats would be really important and useful for readers, mainly clinicians.
Response. The point is taken. I added the diagram (Fig.1) that depicts the thalamocortical system and briefly introduced it in the text. As to the malfunction of this network which could explain cognitive comorbidities, I could suggest the concept of the cognitive thalamus introduced in Section 5.
In order to clarify this issue, I added to page 2: “Dysfunction of thalamocortical network could underlie cognitive comorbidities, as suggested by the concept of thalamocortical dysrhythmia [25,26] and the concept of the cognitive trhalamus (the latter is introduced below, in Section 5).”
Some minor comments:
1.- Line 36. Absence seizures occur without warning but the rest of seizures occur in the same way. Please, modify this sentence.
Response. In view of the fact that absence seizures could occur without warning, they could be extremely frightening for patients and their families.
2.- Please, indicate the references supporting the lines 58-60, because, in fact, this is a very relevant aspect of the manuscript.
Response. Corrected. There are a huge number of studies indicating that “the thalamocortical system generates sleep spindle oscillations and spike-wave discharges (SWDs)” for instance, Steriade 2003;Buzsáki 1991;Polack 2006;Halász 2013;Van Luijtelaar 2006;Halász 2012.
3.- Lines 102-103. Please, clarify the concepts of face and predictive validities.
Response. Ok. I cited Willner’s paper: “Face validity [is] the phenomenological similarities between the model and the condition being modeled… Predictive validity concerns the success of predictions made from the model, and construct validity concerns its theoretical rationale“ [63].
4.- Line 296. Define the acronym NEC.
Response. Corrected.
Round 2
Reviewer 4 Report
Comments and Suggestions for Authors
Thank you for your modifications.